# Proposal of a Machine Learning Approach for Traffic Flow Prediction

**DOI:** 10.3390/s24072348

**Published:** 2024-04-07

**Authors:** Mariaelena Berlotti, Sarah Di Grande, Salvatore Cavalieri

**Affiliations:** Department of Electrical Electronic and Computer Engineering, University of Catania, Viale A. Doria 6, 95125 Catania, Italy; sarah.digrande@phd.unict.it (S.D.G.); salvatore.cavalieri@unict.it (S.C.)

**Keywords:** forecasting, artificial intelligence, traffic congestion, urban scenario, smart city

## Abstract

Rapid global urbanization has led to a growing urban population, posing challenges in transportation management. Persistent issues such as traffic congestion, environmental pollution, and safety risks persist despite attempts to mitigate them, hindering urban progress. This paper focuses on the critical need for accurate traffic flow forecasting, considered one of the main effective solutions for containing traffic congestion in urban scenarios. The challenge of predicting traffic flow is addressed by proposing a two-level machine learning approach. The first level uses an unsupervised clustering model to extract patterns from sensor-generated data, while the second level employs supervised machine learning models. Although the proposed approach requires the availability of data from traffic sensors to realize the training of the machine learning models, it allows traffic flow prediction in urban areas without sensors. In order to verify the prediction capability of the proposed approach, a real urban scenario is considered.

## 1. Introduction

According to recent studies, more than half of the population of the world currently resides in cities and, in a few decades, this percentage is expected to rise [1].

This ever-increasing urban population has led to an exponential rise in the number of vehicles to record highs, putting transport systems under enormous pressure and causing problems such as congestion control, directly affecting citizens through increased travel times, traffic, accidents, and traffic law violations [2].

Despite the many attempts made to mitigate the grave effects of the car industry expansion, traffic congestion, with its associated issues, persists and slows down the development of urban areas.

The prediction of future traffic conditions plays an important role in mitigating traffic congestion issues and their related problems. Machine learning models provide a data-driven approach to traffic forecasting, utilizing historical traffic data to make accurate predictions. Accurate and timely traffic flow prediction is crucial for government agencies, as they can promptly intervene in decisions about traffic management based on the results of traffic flow forecasting [3,4,5,6,7,8,9,10]. Individual travelers can also benefit from traffic flow prediction systems allowing them to adapt their routes inside an urban scenario to future traffic conditions, reducing time wasted in traffic [11,12]. In addition, knowing traffic patterns in advance allows for a strategical allocation of resources (e.g., traffic police, emergency services) to be deployed to areas expected to experience higher traffic volumes. Efficient traffic flow forecasting can enhance environmental impact, reducing emissions, optimizing traffic light timings, and promoting public transportation usage during peak hours. For all these reasons, these tools can lead to significantly enhanced public safety, and reduced environmental impact and provide useful information for urban planners, politicians, and stakeholders.

Traffic flow prediction could be divided into short-term forecasting and long-term forecasting. The former relates to traffic in the near future from a few seconds to a few hours based on current and past traffic information [13,14], while the latter relates to traffic for entire days in the future. Short-term forecasting aids users in optimizing their paths, while long-term forecasting is useful for agencies focused on management and signal control plans for efficient operations [15].

Due to the importance of both kinds of prediction, the authors propose a 24 h forecasting model that is able to predict from the next timestep (1 h) up to 24 h later. Following the recommendation in [16], data were aggregated hourly to reduce the noise impact on the measuring similarity between traffic flow patterns. By evaluating the approach on a real-world case study, the authors want to emphasize the potential of these technologies. The paper presents an extended version of the work developed in [17]. The authors enhanced the earlier research by introducing a more intricate model, trained using one year of data instead of the initial 3-month period. This model is able to detect diverse patterns, identifying not only the daily seasonality across different hours of the day and the weekly seasonality, but also the yearly seasonality. Furthermore, additional experiments were conducted to test the models during holidays such as Christmas, when there are temporal irregularities in the traffic flow.

The remainder of the paper is structured as follows. Section 2 will provide an overview of the state-of-the-art methods regarding this paper’s subject; it ends with the novelty of the proposed approach. In Section 3, the authors provide a detailed explanation of the proposed approach. Section 4 will display the principal outcomes of the proposal testing. Finally, in Section 5, concluding remarks will summarize the contents of this paper.

## 2. Related Work

The aim of this section is to provide an overview of the approaches for traffic flow forecasting that exist in the current literature to point out the differences with the proposal here presented.

In the current literature, traffic flow prediction methods are broadly categorized into three groups [18,19]. The first category comprises statistical methods based on mathematical theory. For instance, [20] proposed a History Average Model for static prediction in urban traffic control systems. Instead, according to [21], the Autoregressive Integrated Moving Average model (ARIMA model) is suitable for predicting stable traffic flows by considering the sequence as a random time sequence. The second category involves machine learning (ML) techniques such as regression analysis [22] and boosting algorithms [23] like LightGBM [24] and CatBoost [25], often used to identify the traits and patterns within historical data progression and to perform predictions. Finally, the third category encompasses deep learning (DL) techniques, particularly neural networks like Back Propagation (BP) [26] and Long Short-Term Memory (LSTM) [27] neural networks. Recently, many works have demonstrated the power of CNNs in a wide variety of tasks [28]. Space graph convolutional networks extend the idea of convolutional neural networks (CNNs) gaining popularity in traffic prediction tasks as they capture the topological structure of transportation networks [29]. In [30], the authors proposed the so-called Phase Space Graph Convolutional Network (PSGCN), an innovative framework based on a graph convolutional network architecture, engineered to model chaotic time series. The uniqueness of the PSGCN lies in its ability to map complex systems into a graph-based representation, that is, the phase space, enabling the analysis of nonlinear dynamics and patterns. This aspect is particularly advantageous for modeling traffic flow data, known for their chaotic nature due to a complex mix of factors such as vehicle speeds, densities, traffic regulations, and the influence of unpredictable events such as accidents and road closures.

Similarly to Spatial CNNs, space visibility graphs are emerging in real-time traffic management and traffic flow prediction applications. Transforming time series into a graph format through a visibility algorithm allows us to model extremely unstable traffic volumes as complex networks and propose deep learning frameworks for network-scale prediction [31].

Traffic flow prediction approaches are generally based on the analysis of input data obtained from traffic sensors installed on urban roads. Typically, these sensors count the number of vehicles passing in front of them, allowing the estimation of relevant traffic flow. The installation of traffic sensors on all roads of an urban scenario is not possible due to costs and other logistical reasons. Consequently, many current approaches utilize data collected by sensors on a subset of urban roads to predict traffic flow not only on roads where sensors are installed, but also on others where data collection is not feasible. This is often achieved by examining spatio-temporal characteristics between neighboring and distant sensors to predict traffic flow in urban areas similar to the ones of the collected data [32]. For example, [33] introduced a spatio-temporal traffic flow estimation model that utilizes data from multiple locations within the network. The approach incorporates various features beyond solely relying on traffic flow data.

The analysis of traffic sensor data considering other information different from the trivial vehicle counts requires a lot of effort; moreover, this extra information may be too tied to a specific urban scenario (e.g., the road paving and size), making the traffic flow prediction approach difficult to be extended to different scenarios. For these reasons, many current approaches limit the analysis of data to information obtained only from traffic sensors. Among the existing solutions, the analysis of data is realized by classifying sensor data using machine learning clustering approaches. Sometimes, a two-level machine learning method is adopted; an unsupervised clustering model organizes sensor data into clusters (grouping data sharing similar features), while a supervised machine learning model predicts traffic flow for each cluster.

Among two-level data-driven approaches present in the literature, the most recent is presented in [34]; the authors of this paper proposed a traffic flow method based on the application of a clustering algorithm for the short-term forecasting of traffic flow in the highways of Seattle. In their scenario, only one sensor was considered to be producing traffic flow information. The traffic flow time series for a single sensor was segmented into 24 h intervals, given as input to the clustering algorithm to identify days with similar patterns. As a result of the clustering procedure, four groups were created. Four different predictors were trained, one for each group. These predictors will always look at the target day to predict 24 h, meaning that they are specialized for a single daily pattern. Consequently, if predictors are tested with a day experiencing a significantly different traffic flow pattern from the four target days, the algorithm will not make accurate predictions.

In this paper, the authors propose a two-level machine learning methodology that involves the use of clustering and forecasting models. Differently from the researchers of the approach discussed before (i.e., that described in [34]), the authors applied a clustering methodology to group roads with similar patterns using one year of data for several sensors as input to the algorithm. Furthermore, the sensor data used in this work refer to heterogeneous roads with different physical and traffic flow characteristics, including not just highways, but also roads located in downtown and suburban areas.

After clustering the roads, a predictor was trained with one year of data for each cluster. Differently from [34], the authors tested each model on a time series excluded from the training set, exhibiting a similar pattern to the ones present in the clusters. Additionally, the approach was evaluated on a greater number of sensors using two weeks of data as a test set, including working days, weekends, and holidays.

Another difference from the work in [34] concerns the type of prediction; that study focused on short-term forecasting, whereas the current paper proposes a combination of short- and long-term forecasts, as highlighted in the Introduction and further detailed in the remainder of the paper.

In conclusion, this section pointed out that the proposed approach significantly differs from existing methodologies found in the current literature. Section 3 will delve into a detailed description of the approach.

## 3. Materials and Methods

In this section, the authors delve into the materials and methods used to tackle the specified forecasting problem when employing a two-level machine learning approach. To begin, since real-world data were used to test the proposed approach, detailed descriptions of the urban scenario from where data were collected and its relevant traffic patterns will be provided. Additionally, the preprocessing steps taken to prepare the final dataset before the construction of the machine learning models will be elucidated. Finally, a comprehensive description of the proposed approach will be outlined in detail.

### 3.1. Case Study Description

The traffic flow prediction approach presented here is based on real data obtained from a network of traffic sensors located in Catania, Italy.

The city forms the core of a larger metropolitan area that includes the primary municipality of Catania along with 26 adjacent urban centers. The transportation demand of the city primarily depends on private vehicles, influenced by the lifestyle choices and preferences of its residents. This reliance on private transport is due to the concentration of economic and residential activities within the main boundaries of the city.

Over time, the population of Catania has expanded beyond the city limits, resulting in the formation of an integrated urban network that extends beyond municipal boundaries. This expansion has significantly increased traffic influx from the surrounding suburbs and neighboring municipalities, leading to daily congestion in the city center and heightened levels of environmental pollution. This surge in transportation demand has substantially augmented individual vehicle ownership, further burdening an already stressed urban infrastructure. Consequently, there has been a marked escalation in congestion, underscoring the urgent necessity for an effective management of traffic congestion in Catania through the adoption of traffic flow forecasting methodologies.

### 3.2. Data Collection and Preprocessing

Data about traffic flow were collected through sensors, specifically 21 microwave traffic counters known as MobilTraf300 manufactured by FAMAS [35], placed across the Catania urban area. These traffic counters harness the most recent technologies in 24 GHz radar frequency microwaves, signal analysis, and both local and remote communications fields. Furthermore, these stand-alone devices integrate sensors, control electronics, and a hybrid power system consisting of batteries and photovoltaic panels to operate with minimal energy consumption while delivering precise measurements.

Different kinds of information are collected using traffic counters. When a vehicle passes through the electromagnetic field produced by two MobilTraf300 sensors, these units capture various vehicle-related data, including the date and time of passage, travel direction, and the specific lane used for transit.

To retrieve this information, MobilTraf MANAGER, a traffic manager software developed by FAMAS, was utilized. Data from each traffic sensor are obtained in 5 min intervals for the specified timeframe. For each timestamp, the number of vehicles passed in the last 5 min is collected.

Twelve operational traffic counters (TCs) were selected from the available ones at the time of data retrieval. The analysis period covers 1 January 2022 to 31 December 2022.

Each TC corresponds to a specific road, displaying distinct characteristics. These roads are categorized as follows:Single-lane roads;Two-lane roads in the same direction;Two-lane roads in opposite directions.

For single-lane roads, the downloaded time series contains ‘Timestamp’ and ‘Traffic Flow’ columns. Table 1 reports an example of a dataset for the single-lane road. In the dataset, the ‘Traffic Flow’ column represents the total traffic flow for each timestamp.

For two-lane roads, the dataset maintains the same structure for both cases of lanes with the same and opposite directions, including timestamps, traffic flow in the first lane, and traffic flow in the second lane. Table 2 provides an example of this type of dataset for a two-lane road, with two distinct columns representing the traffic flows in the two lanes.

To prepare a well-structured dataset suitable for traffic flow prediction using machine learning algorithms, distinct data preprocessing steps were performed, aiming to organize data according to the template shown in Table 3. For roads featuring a single lane, Table 3 includes one column representing the traffic flow for that road. For roads with two lanes in the same direction, Table 3 presents a single column that sums the traffic flows from both lanes of the same road. For roads with two lanes in opposite directions, Table 3 has two adjacent columns, each corresponding to the traffic flow in one of the two lanes of the same road.

The timestamps shown in Table 3 match those in the various tables containing traffic flow data for each road. As mentioned, these timestamps correspond to 5 min intervals, reflecting that data from each traffic sensor are collected in 5 min periods over the specified timeframe.

As said in the Introduction, in this study, the authors adopted both short-term and long-term forecasting approaches. In detail, the model forecasts traffic flow for the next hour up to the next 24 h. According to this assumption, the traffic flow information is aggregated at a time interval of 1 h instead of the original 5 min interval. Subsequently, the first operation conducted on the dataset, with a structure as depicted in Table 3, was aggregation; in particular, the traffic flow information present in each column of Table 3, was summed inside each time interval of 1 h. Due to this aggregation, timestamps correspond to 1 h intervals.

The treatment of outliers and missing values within the dataset was executed as follows. Outliers were identified using a boxplot and subsequently removed from the dataset. This method facilitated the recognition of aberrant data points that significantly deviated from the typical value range present in the dataset. Addressing these outliers was imperative due to their potential to distort subsequent analysis or modeling outcomes [36,37]. Hence, upon visual identification using the boxplot, these outliers were eliminated from the dataset to uphold the integrity and accuracy of subsequent procedures. Besides the removed outliers, additional missing values were found in the dataset. These missing values were attributed to sensor malfunctions. The malfunctioning sensors resulted in missing observations when a particular TC ceased to function, failing to collect traffic information for specific dates and times. To handle missing observations, the selected approach entailed filling in missing values using a time-based averaging technique. This method computes the average traffic values for the corresponding road, day of the week, and time within the same month. Utilizing this technique ensures data integrity while addressing missing data points in a meaningful and statistically informed manner.

The final dataset comprised a timestamp column with 1 h intervals and 14 columns representing the total vehicle count for each lane at all hours of the day across 365 days; the total number of rows was 8769.

Before proceeding with the development of machine learning models, it was necessary to perform data normalization due to the significant diversity in the value ranges of the considered time series. In this analysis, Min–Max scaling was utilized as the normalization technique, which standardizes the range of each variable to a uniform 0–1 scale. Indeed, this normalization technique was applied to each variable in the dataset independently. The formula used to calculate normalization is the following, where *x* = (*x*_1_, …, *x_n_*) is a variable with different observations and *z_i_* is the normalized value:(1)$$z_{i}=\frac{x_{i}-min(x)}{\max\left(x\right)-min(x)}$$

The formula computes the normalized value for a specific data point by subtracting the minimum value of the variable from the data point and then dividing this difference by the range of values. This process scales the data proportionally, ensuring that the minimum value maps to 0 and the maximum value maps to 1 while retaining the relationship between all other values within the range.

### 3.3. Overview of the Proposed Machine Learning Approach

As mentioned previously, the main objective of this study involved employing a two-level machine learning methodology that integrates both unsupervised and supervised models. First, an unsupervised model was employed to analyze the time series data collected from the sensors, which depict traffic flow. This involves using clustering techniques to group similar data and unveiling unseen patterns, relationships, or structures within the dataset. These discovered patterns are then organized into distinct clusters based on their similarities. More detailed information about the clustering technique will be provided in Section 3.3.1.

Afterward, a supervised machine learning model is created for each of these clusters. These models are specifically designed to predict traffic flow for data within their respective clusters. To decide which cluster a new road with limited available data belongs to, distance metrics are used to find the closest match among the existing clusters.

Once the appropriate cluster for the new road segment is identified, a machine learning model that was previously trained on traffic flow data from similar roads within the same cluster is applied to forecast traffic flow for this new segment. This method results in the development of multiple distinct models, each focused on forecasting traffic flow for roads that have limited observations but sharing similar patterns within their respective clusters. More details about the forecasting approach will be provided in Section 3.3.2.

Figure 1 presents a visual representation of the proposed two-level approach to traffic flow forecasting. Starting on the left, *t* denotes the yearly time series of traffic flow. More precisely, traffic data aggregated on an hourly basis over an entire year from each road segment were utilized as input for a clustering model. This model generated a set of clusters, (e.g., C_1_, C_2_, C_3_, and C_4_, as shown), with group roads exhibiting similar traffic flow patterns. Subsequently, a distinct forecasting model is trained for each cluster, utilizing the subset of time series data corresponding to the roads within that cluster. Finally, every forecasting model—each calibrated to a specific cluster—aims to predict the traffic flow for the forthcoming 24 h period for each subset of the time series within its cluster.

The findings presented in this paper highlight that each model, trained extensively on time series data within the same cluster, can proficiently generate forecasts for similar yet unseen series during its training, demanding only a small number of observations from these new series. In instances where roads lack sensors, essential traffic data for the forecasting models can be obtained from various sources like Floating Car Data (FCD). Therefore, in scenarios where sensors are unavailable for a particular road, forecasting for that road can be accomplished using a small subset of data acquired from alternative sources. The use of FCD is outside the scope of this paper.

#### 3.3.1. Clustering Step

In this study, clustering was applied to time series data. Clustering this type of data involves grouping similar temporal sequences or patterns within datasets. Unlike traditional clustering applied to static data, clustering for time series considers the sequential nature of data points and aims to identify similar trends, patterns, or behaviors over time.

The clustering algorithm utilized in this study, known as Time Series K-Means (TSkmeans), represents an adaptation of the traditional K-means algorithm specifically designed for clustering time series data [38,39].

TSkmeans diverges from the conventional K-means applied to static data by incorporating not only the values of data points, but also their temporal relationships when forming clusters. Unlike standard K-means, which primarily considers the magnitude of data points, TSkmeans places significant emphasis on the sequential order and patterns exhibited over time within the data.

The presented clustering algorithm involves the utilization of the Dynamic Time Warping (DTW) metric for assessing similarity among temporal sequences. Unlike the typical Euclidean distance utilized in standard K-means, DTW is a specialized distance metric specifically tailored for time series data. It accounts for temporal shifts, varying speeds, and local distortions present in sequences, thus providing a more comprehensive measure of similarity between time-dependent data points.

By incorporating DTW into the clustering process, TSkmeans offers a more nuanced understanding of similarities and dissimilarities within time series data. This integration of temporal considerations enhances the abilities of the algorithm to capture and identify patterns that might otherwise be overlooked by traditional clustering techniques, thereby enabling more accurate and meaningful cluster formations in time series analysis.

The initial phase of clustering involves the critical task of determining the optimal number of clusters (K). To accomplish this, a silhouette score was employed as a metric. The silhouette score serves as a statistical measure to evaluate the overall quality of clustering by considering two essential aspects: the degree of separation between data points within the same cluster and the degree of separation between data points assigned to different clusters.

For each data point, the silhouette score is computed considering the intra-cluster distance—the mean distance between the data point and all the other points in the same cluster—and the inter-cluster distance—the mean distance between the data point and all the other points in the nearest neighboring cluster. The overall silhouette score of the clustering result is calculated by averaging the individual silhouette score of all the points.

This scoring system operates within a range from −1 to +1. A silhouette score of +1 signifies an ideal scenario where clusters are perfectly distinct and well separated, indicating robust and accurate clustering. Conversely, a score of −1 indicates suboptimal clustering, revealing a considerable overlap or misallocation of data points among clusters, signifying poor clustering quality. Moreover, a score close to 0 suggests ambiguity or uncertainty in the assignment of data points to clusters, indicating challenges in defining clear boundaries between clusters. In utilizing the silhouette score, the number of clusters that obtained the highest cluster score was selected to develop the final model, thus ensuring well-defined and distinct clusters for the given dataset.

#### 3.3.2. Traffic Flow Forecasting Step

Different sub-steps were performed for the forecasting part. Initially, the dataset was divided into two segments: a training set and a test set. The training set encompassed observations from 1 January 2022 to 16 December 2022. In contrast, the test set comprised time series data spanning 17 December 2022 to 31 December 2022. All the machine learning models used for forecasting purposes in this study were implemented using Darts, a specialized Python library for time series data analysis, particularly focused on time series prediction. Darts offers an intuitive API design and a comprehensive suite of contemporary machine learning tools [40]. The following steps were iterated using four distinct algorithms to determine the optimal one: CatBoost, LightGBM, Random Forest, and XGBoost. Further details of these machine learning algorithms will be provided in Section 4.

The overarching objective of the forecasting process was to predict the traffic flow for a specific road direction by employing a model trained on time series exhibiting comparable patterns. This model was designed to generate forecasts based on a minimal set of observations. Consequently, despite training the model with a substantial dataset spanning a year collected from sensors distributed in the city, it has the capability to provide forecasts for similar series with a limited dataset downloaded from alternative sources. This robustness in training enables the model to effectively anticipate traffic flow patterns despite the reduced amount of input data, showcasing its adaptability and efficiency in forecasting for comparable scenarios.

To thoroughly evaluate the effectiveness of the approach, several experiments were undertaken. Specifically, for every cluster, multiple iterations were carried out, each involving the creation of various models. During each iteration, a distinct time series was systematically removed from the training data, and the model performances were assessed based on the model’s ability to forecast this excluded time series. This process simulated scenarios where a particular series had a limited number of available data points for which accurate forecasts were needed. In systematically excluding different time series and evaluating the predictive capabilities of the models in such scenarios, the robustness of the approach in handling situations with sparse or limited data was thoroughly tested and assessed.

For instance, in a cluster C_x_ comprising n_x_ time series, the methodology was tested by repeating the process n_x_ − 1 times. Each iteration utilized n_x_ − 1 out of the n_x_ time series for training the model and reserved the last one for testing. This cycle was iterated n_x_ times in total, ensuring that each time series was excluded once from the training data and exclusively used for testing. Thus, when considering, for example, three clusters C_1_, C_2_, and C_3_, with, respectively, n_1_, n_2_, and n_3_ series, the total number of models developed and tested is the sum of n_1_, n_2_, and n_3_.

Each model was tested for four different machine learning algorithms, as explained before, and different sets of hyperparameters. These hyperparameters, crucially chosen before model training, significantly impact model performance. To optimize these parameters, the Optuna 3.6.1 Python library was employed [41].

The chosen objective function for optimization across the training set was the validation loss. This function measures how well a model performs on a separate validation dataset during the process of optimizing hyperparameters. Specifically, it quantifies the error or deviation between the model predictions and the actual values within this validation dataset. To carry out this evaluation, the last 24 h of the training set were set aside as the validation window.

Each time a model was created, for each combination of algorithm and set of hyperparameters, 100 different trials were conducted to find the best configuration of hyperparameters for the selected algorithm and the series used for model development.

The ML algorithms implemented in the forecasting process, used in this proposal are the following: LightGBM version 4.1.0, XGBoost version 2.0.2, CatBoost version 1.2.2, and Random Forest from scikit-learn 1.4.0 Python library. In Section 4, the authors provide an overview of these ML algorithms.

During forecasting, the models’ performances were evaluated through a walk-forward validation method. Unlike standard cross-validation methods that involve random splits, walk-forward validation preserves the temporal order of the data. This technique is well suited for identifying time-dependent patterns in time series data [42,43,44].

The validation process starts by initially training the model using historical time series data ranging from January to 16 December. Once trained, the model makes predictions for the next 24 consecutive time steps in the sequence. These predictions are compared against the actual observed values to evaluate the initial performance of the models.

After this initial phase, the training data window is shifted forward by 24 steps. This prompts the model to be retrained using the updated dataset, incorporating new information.

Subsequently, the model generates predictions for the upcoming time steps based on these new training data. These predictions are then compared with the actual observed values to assess the model performance in this updated timeframe.

This iterative process continues as the window progresses through the entire time series. With each shift of the training data window, the model is repeatedly trained and tested on new data, allowing for the continuous evaluation and refinement of its predictive accuracy throughout the entire time series. This iterative approach enables the model to adapt to evolving patterns or changes in the data distribution, enhancing its ability to make accurate predictions across different time periods within the series.

The walk-forward validation method closely simulates a model behavior in a dynamic, real-world setting. Regular retraining allows the model to adapt dynamically to changes in data distributions or patterns over time, significantly enhancing its practical effectiveness.

The metrics used to compare predicted and actual values were the mean absolute error (MAE), the symmetric mean absolute percentage error (SMAPE), the mean square error (MSE), and the root mean square error (RMSE) [45,46,47,48]. The mean absolute error (MAE) calculates the average of the absolute errors in the predictions, quantifying the disparity between predicted and actual values. The symmetric mean absolute percentage error (SMAPE) measures accuracy through relative errors expressed as a percentage. The mean square error (MSE) determines the average squared deviation between predicted and actual values. The root mean square error (RMSE) represents the square root of the mean of squared errors, functioning as an additional performance metric used to assess prediction accuracy. Lower values for each metric indicate a better model performance. It is essential to highlight that while SMAPE is the primary performance metric used to select the best model, the other measures, MAE, MSE, and RMSE, are also considered supplementary indicators during the evaluation process.

## 4. Overview of Machine Learning Models

In this section, the authors provide overviews of the machine learning models implemented in the forecasting process, providing a clearer understanding of their roles and how they impact the model’s learning and performance.

LightGBM, short for Light Gradient Boosting Machine, is a gradient-boosting framework that uses tree-based learning algorithms. Proposed by [24] to solve the issue of the Gradient Boosting Decision Tree (GBDT) in conventional implementations, it is based on two novel techniques, Gradient-Based One-Side Sampling (GOSS) and Exclusive Feature Bundling (EFB) [49]. In [24], the experiment demonstrated that LigthGBM can accelerate the training process by up to over 20 times. In addition, the model grows trees leaf-wise (vertically) rather than level-wise (horizontally) which leads to a better reduction in loss and, hence, generally, more accurate predictions. Some hyperparameters directly influence the complexity, speed, and performance of the model; brief explanations are provided as follows:n_estimators: common in ensemble methods like LightGBM, XGBoost, and Random Forest, it specifies the number of trees to build. More trees can capture more complex patterns but also increase the risk of overfitting and increase the computational costs.learning_rate: found in gradient boosting models like LightGBM, XGBoost, and CatBoost, this parameter controls the rate at which the model adapts to the problem over each boosting round. A smaller learning rate requires a higher number of boosting rounds (n_estimators) to be performed by the model. Tuning this parameter is crucial for controlling overfitting and the convergence speed of the model.num_leaves: as the most critical parameter in LightGBM, it controls the complexity of the tree model. A higher number of leaves allows the model to learn finer details but also makes it more susceptible to overfitting. The optimal number of leaves needs to be established by also looking at the size of the dataset.max_depth: this parameter sets the maximum depth of a tree. It is used to control overfitting as deeper trees can learn more complex patterns but might also learn noise in the data.min_child_samples: specifies the minimum number of samples (data points) needed to create a new node in a tree. A higher number makes the model more conservative, preventing it from learning from very small, potentially noisy groups.

XGBoost, or eXtreme Gradient Boosting [23], is another ML algorithm belonging to the gradient boosting framework. As it has been widely used by data scientists in many machine learning and data mining challenges, the key factor contributing to the success of XGBoost lies in its scalability across various scenarios (i.e., see [50,51]). Operating over ten times faster than existing popular solutions on a singular machine, the algorithm achieves this great scalability thanks to the combination of crucial system and algorithmic optimizations. Notable innovations of the XGBoost algorithm include Exact Greedy Algorithm, Weighted Quantile Sketch, and Sparsity-Aware Split Finding [52]. Like LightGBM, the XGBoost algorithm features the hyperparameters n_estimators, learning_rate, and max_depth. In addition, the following hyperparameters directly influence the performance of the ML algorithm:min_child_weight: as the minimum sum of instance weight needed in a child, this parameter is like min_child_samples in LightGBM and helps control overfitting.gamma: serving as a regularization parameter, it specifies the minimum loss reduction required to make a further partition on a leaf node of the tree.reg_aplha and reg_lambda: corresponding to the L1 (Lasso regression) and L2 (Ridge regression) regularization terms, these add penalties to the magnitude of the coefficients to prevent overfitting by keeping the weights as small as possible.

The CatBoost algorithm is a Gradient Boosting Decision Tree (GDBT) framework based on a symmetric decision tree. Proposed by [53], in recent years, this algorithm turned out to be particularly effective for both classification and regression tasks. The main idea of ensemble learning methods like the CatBoost algorithm is to generate a stronger predictive model by combining weak learners as decision trees. Indeed, the algorithm processes a series of simple decision tree models sequentially to optimize their performances by reducing the errors made by the previously trained models. Further details on the CatBoost algorithm are provide in [54]. Below, the hyperparameters involved in the CatBoost algorithm are described:iterations: this is the number of trees that the algorithm builds before stopping. More iterations allow the model to fit the data better but can lead to overfitting.l2_leaf_reg: used to prevent overfitting by penalizing large weights, this parameter represents the coefficient for L2 regularization.random_strength: this parameter is used for adding randomness to the objective function. It makes the model more robust by providing a smoother decision boundary.

In addition to these specific hyperparameters of the CatBoost algorithm, learning_rate and depth, which are described above, must be optimized in order to improve the performance of the algorithm.

Finally, the Random Forest algorithm, proposed by [55], is an ensemble learning technique that is exceedingly used for both classification and regression tasks. The approach combines multiple decision trees during training, and it outputs the average prediction (for regression) or the mode prediction (for classification) of the individual trees. The key idea behind Random Forest is to introduce randomness into the training process to improve the overall performance and the robustness of the model [56]. Specific hyperparameters of the Random Forest algorithm are as follows:min_samples_split: this is the minimum number of samples required to split an internal node. Higher values prevent the model from learning relations that might be highly specific to the sample selected for a tree.min_samples_leaf: this is the minimum number of samples required to be at a leaf node. Setting this parameter helps in smoothing the model, especially in regression tasks.

## 5. Results and Discussions

As a result of the clustering process, a specific cluster configuration was obtained, yielding a silhouette score of 0.52. This outcome is considered quite favorable, suggesting that the clusters are adequately separated, though not entirely distinct. It is also noteworthy that a team of experts in the field verified the efficacy of the clustering algorithm in grouping roads with similar structural characteristics.

Figure 2 illustrates the results of the clustering step. In the graph, every curve corresponds to the time series data recorded by a sensor placed on a particular road in Catania. Due to limitations in space, the figure shows up to just one week of data even if the algorithm was run with an entire year of data as input.

The resulting clustering configuration shows a first cluster made of five time series, a second cluster consisting of three time series, and a third cluster encompassing five time series.

As it consisted of only one time series, the fourth clustering was excluded from the figure, because of its singular nature.

In keeping in mind the outcomes of the clustering step, forecasting was implemented. Firstly, the authors tested the set of optimized machine learning models described in Section 4.

The dataset was divided into two sets: a training set including data from 1 January 2022 to 16 December 2022 and a test set consisting of two weeks of observations, spanning from 17 December 2022 to 31 December 2022. This choice was influenced by the fact that in Italy, the last week of December is Christmas week. The traffic patterns in Catania during this period deviate from the usual, particularly impacting certain roads that are notably affected by the onset of the Christmas holidays. For instance, the traffic recorded by the MT9 sensor predominantly involved private vehicles heading toward the University of Catania. With the closure of universities during Christmas, there is a marked alteration in the traffic dynamics along this road.

In the optimization framework, two hyperparameters had to be configured: input_chunk_length and output_chunk_length. The input_chunk_length represents the amount of time steps (in hours) that the model utilized to perform predictions. Conversely, the output_chunck_length is the number of time steps (in hours) predicted at once by the internal model. If the input_chunk_length was optimized, the output_chunck_length was set to 24 h. The decision to set the prediction step size to the next 24 h was dictated by the authors’ motivation to present their model as a useful tool for urban planning. Essentially, long-term forecasting provides a comprehensive projection of traffic flow patterns to assess future capacity needs, allowing for more precise urban planning and improved infrastructure development decisions [57]. Meanwhile, real-time traffic management systems require predictions at shorter intervals [58].

The proposed models utilize a predetermined number of hours (input chunk length) to forecast the upcoming 24 h.

The hyperparameters of the optimized models are listed as follows:LightGBM: {input_chunk_length = 354, num_leaves = 85, max_depth = 1, learning_rate = 0.1620, n_estimators = 23, min_child_samples = 35}.XGBoost: {input_chunk_length = 224, n_estimators = 30, learning_rate = 0.1980, max_depth = 6, min_child_weight = 4, gamma = 3, reg_alpha = 0.3216, reg_lambda = 0.9846}.CatBoost: {input_chunk_length = 287, iterations = 989, learning_rate = 0.1559, depth = 5, l2_leaf_reg = 3, random_strength = 7}.Random Forest: {input_chunk_length = 263, n_estimators = 58, max_depth = 10, min_samples_split = 3, min_samples_leaf = 5}.

In Table 4 and Table 5, the authors report the average performance metrics obtained from testing the models on data from all 14 sensors for both the week from 17 December to 23 December 2022 and on the Christmas week from 24 December to 31 December 2022, respectively.

In analyzing the performance metrics, the XGBoost and CatBoost outperformed LightGBM and Random Forest across all metrics. This consistency suggests the robustness and reliability of these boosting algorithms in handling the forecasting task compared to the other models.

Focusing on the comparison between XGBoost and CatBoost, as reported in Table 4, they exhibited very similar performances during the week from 17 December to 23 December, suggesting that they are both effective choices for the task. However, while there was little difference in performance between these two models during this tested week, XGBoost demonstrated superiority over CatBoost during the Christmas week. This result suggests that the XGBoost model is more robust than CatBoost when variations in the normal traffic flow patterns are introduced. Based on these considerations, the third best-performing model utilized the LightGBM algorithm, and the fourth best-performing model employed the Random Forest algorithm. Based on these results, the XGBoost algorithm was used for the short-term forecasting task.

To test the approach, the model was trained on all the available time series belonging to a cluster, excluding data from a different sensor each time. In other words, this process involved creating different XGBoost models repeatedly, leaving out one specific time series from the training data each time and then evaluating the performance of the model based on the omitted time series.

Below, the authors listed the XGBoost models obtained by excluding a different time series from the first cluster each time:First model: {input_chunk_length = 214, n_estimators = 100, learning_rate = 0.6171, max_depth = 1, min_child_weight = 5, gamma = 2, reg_alpha = 0.8155, reg_lambda = 0.4019}.Second model: {input_chunk_length = 239, n_estimators = 65, learning_rate = 0.2549, max_depth = 6, min_child_weight = 8, gamma = 1, reg_alpha = 0.0128, reg_lambda = 0.3413}.Third model: {input_chunk_length = 198, n_estimators = 32, learning_rate = 0.8024, max_depth = 0, min_child_weight = 8, gamma = 5, reg_alpha = 0.6796, reg_lambda = 0.0242}.Fourth model: {input_chunk_length = 362, n_estimators = 61, learning_rate = 0.7500, max_depth = 0, min_child_weight = 3, gamma = 4, reg_alpha = 0.9530, reg_lambda = 0.0577}.Fifth model: {input_chunk_length = 202, n_estimators = 53, learning_rate = 0.6013, max_depth = 4, min_child_weight = 2, gamma = 2, reg_alpha = 0.4324, reg_lambda = 0.3886}.Next, the best hyperparameters for the second cluster are listed as follows:First model: {input_chunk_length = 294, n_estimators = 48, learning_rate = 0.9116, max_depth = 5, min_child_weight = 8, gamma = 2, reg_alpha = 0.7541, reg_lambda = 0.1687}.Second model: {input_chunk_length = 242, n_estimators = 21, learning_rate = 0.8160, max_depth = 5, min_child_weight = 10, gamma = 5, reg_alpha = 0.2143, reg_lambda = 0.2808}.Third model: {input_chunk_length = 247, n_estimators = 60, learning_rate = 0.7606, max_depth = 0, min_child_weight = 8, gamma = 2, reg_alpha = 0.2381, reg_lambda = 0.7919}.Finally, the same procedure was applied for the third cluster, for which the following best hyperparameters were obtained:First model: {input_chunk_length = 201, n_estimators = 22, learning_rate = 0.8273, max_depth = 1, min_child_weight = 9, gamma = 1, reg_alpha = 0.3290, reg_lambda = 0.4874}.Second model: {input_chunk_length = 207, n_estimators = 70, learning_rate = 0.1490, max_depth = 9, min_child_weight = 3, gamma = 53 reg_alpha = 0.8095, reg_lambda = 0.9837}.Third model: {input_chunk_length = 192, n_estimators = 21, learning_rate = 0.9600, max_depth = 6, min_child_weight = 5, gamma = 2, reg_alpha = 0.7949, reg_lambda = 0.3499}.

As explained before, the ‘input_chunk_length’ parameter signifies the duration, in hours, that the model requires to comprehend patterns and generate forecasts for the subsequent 24 h. Upon examining each model, the longest duration needed was 362 h, which is equivalent to approximately 15 days. Consequently, the model necessitates only a minimal number of observations during the forecasting phase. In other words, a small number of observations from unseen roads are needed to forecast these new roads.

Table 6 and Table 7 show the effectiveness of the models in predicting the traffic volume for each sensor of every cluster for the last two weeks of December. These tables report the average performance metrics of the roads included in the training.

Table 8 and Table 9 report the performance metrics computed each time a road was excluded from the three clusters. Also, in this case, the test was conducted on both the week spanning from 17 to 23 December 2022 and on the Christmas week.

Examining the performance metrics in the two tables reveals that the predictions for the week of 17 to 23 December across the three clusters achieved good accuracy. The authors obtained an average error rate (SMAPE) of 15.5585 for Cluster 1, 33.2928 for Cluster 2, and 28.8474 for Cluster 3. As concerns predictions for the week of 24 to 31 December, an overall decrease can be noticed in terms of performance in the majority of the sensors. This can be justified by the abnormal pattern that occurs in the streets of Catania during the Christmas holidays. Finally, a note must be made about the performance of sensor MT9b; indeed, for this sensor, the error rate was quite elevated. The traffic flow of this road is strictly characterized by private and public vehicles moving to universities. To the best knowledge of the authors, in the two tested weeks from 17 to 31 December, universities in Catania closed for the Christmas holidays.

Figure 3 presents the actual and predicted values of traffic flow for some time series excluded by each cluster on the two test weeks spanning from 17 to 23 December 2022 and on Christmas week.

As can be seen from the graphs, for these three tested excluded sensors, the model was able to predict, quite accurately, the traffic flow registered in the week from 17 to 23 December 2022. Moreover, as concerns the holiday week, for the majority of the sensors, the model accuracy slightly decreased when predicting the traffic flow on Christmas and Boxing day (25 and 26 December 2022).

## 6. Conclusions

In this study, the authors proposed a two-level machine learning methodology for forecasting traffic flow data. In detail, they tried to overcome the problems of predicting traffic flow for roads where there are no sensor data available. In using Catania as a reference city for its complex transportation network, a TSkmeans algorithm was used to group data sharing similar patterns, obtaining a final silhouette score of 0.52. After the clusters were detected, an optimized XGBoost model was created, one for each group of data. This model resulted as the best performing and most robust one, able to efficiently predict traffic on roads exhibiting a different pattern from the normal one. Unlike in [28], an extended year of data was provided as input to the model to overcome the limit of having too specialized algorithms for single daily patterns. Moreover, the authors used data from 14 sensors referring to different roads located in the center and suburban areas.

Ultimately, the methodology involved training the model on the entire set of time series within a cluster with the exclusion of a different sensor in each iteration. To clarify, this entails iteratively building distinct XGBoost models by omitting a specific time series from the training data each time and subsequently assessing the model performances by forecasting the traffic flow of the excluded time series. This test aimed to evaluate the capability of the models to generalize on series that exhibit similar patterns to those used during training but were not part of the training dataset. The results indicate a positive outcome, demonstrating the ability of the models to forecast traffic flows on series not encountered during training, even with a limited number of input days (e.g., two weeks).

Future plans involve increasing the number of sensors to cover a large urban area under study. Furthermore, the authors expect to gather diverse data from various sources and to include them in the model (e.g., weather conditions, traffic congestion, roadwork information).

## Figures and Tables

**Figure 1 sensors-24-02348-f001:**
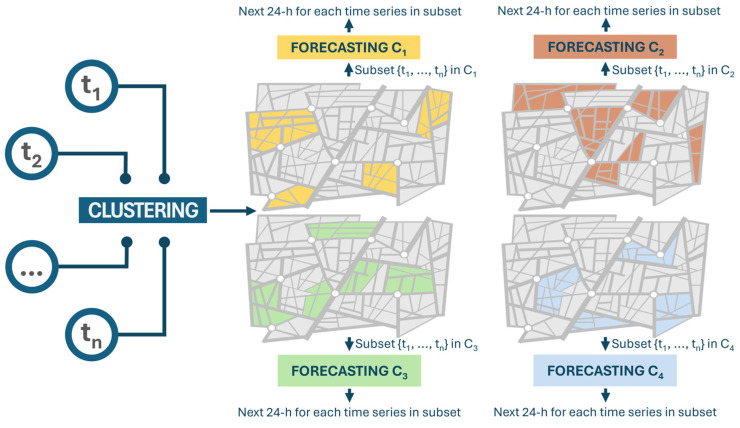
Two-level ML approach for traffic flow prediction.

**Figure 2 sensors-24-02348-f002:**
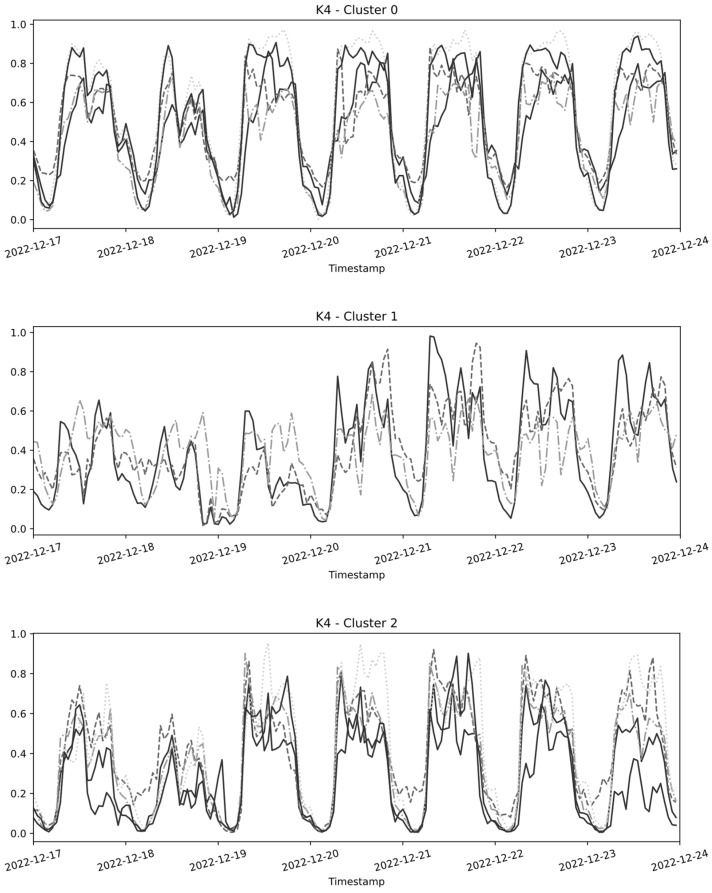
Time series data distributed in the three clusters.

**Figure 3 sensors-24-02348-f003:**
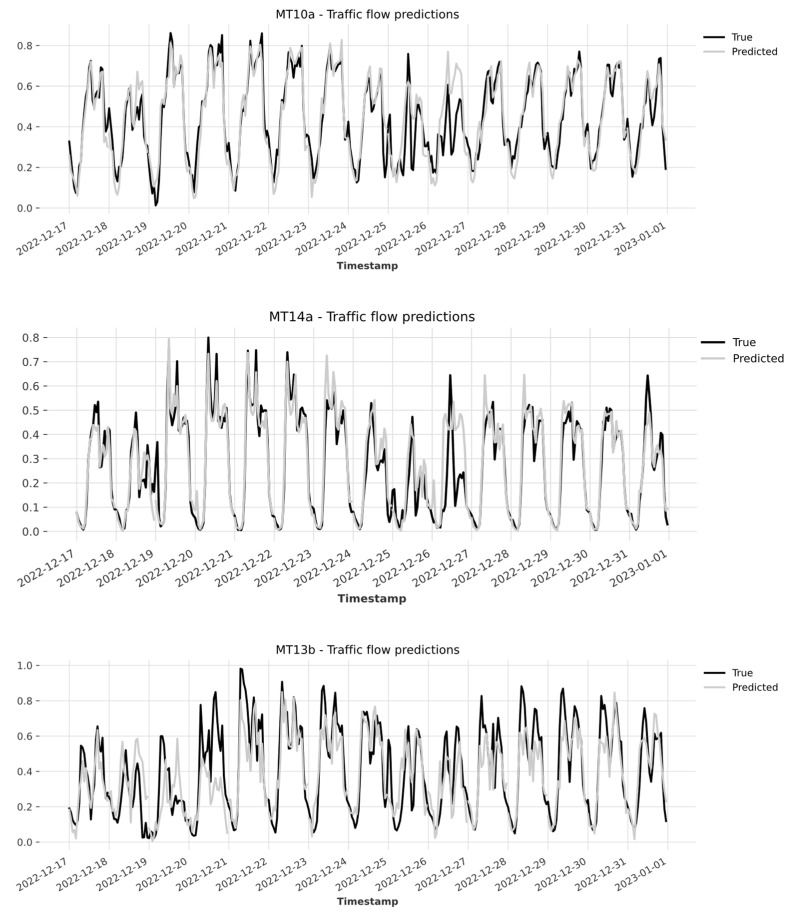
Example test results on excluded time series for the two weeks.

**Table 1 sensors-24-02348-t001:** Example of traffic flow dataset for a single-lane road.

Timestamp	Traffic Flow
2022-01-01 00:00:00	tf_0_
…	…
2022-12-31 23:55:00	tf_n_

**Table 2 sensors-24-02348-t002:** Example of traffic flow dataset for a two-lane road.

Timestamp	Traffic Flow 1° Lane	Traffic Flow 2° Lane
2022-01-01 00:00:00	tfL_0,1_	tfL_0,2_
…	…	…
2022-12-31 23:55:00	tfL_n,1_	tfL_n,2_

**Table 3 sensors-24-02348-t003:** Example of traffic flow dataset obtained after preprocessing steps.

Timestamp	Traffic Flow_id_1	Traffic Flow_id_2	…	Traffic Flow_id_m
2022-01-01 00:00:00	tfL_0,1_	tfL_0,2_	…	tfL_0,m_
…	…	…	…	…
2022-12-31 23:00:00	tfL_n,1_	tfL_n,2_	…	tfL_n,m_

**Table 4 sensors-24-02348-t004:** Performance metrics of optimized models on the week of 17 December to 23 December.

Models	MAE	SMAPE	MSE	RMSE
LightGBM	0.0679	23.3168	0.0103	0.0204
XGBoost	0.0649	21.6715	0.0098	0.0198
CatBoost	0.0637	21.8361	0.0092	0.0185
Random Forest	0.0792	27.0989	0.01259	0.0251

**Table 5 sensors-24-02348-t005:** Performance metrics of optimized models on Christmas week.

Models	MAE	SMAPE	MSE	RMSE
LightGBM	0.0655	27.2436	0.0098	0.0343
XGBoost	0.0632	23.6044	0.0095	0.0327
CatBoost	0.0710	26.8008	0.0109	0.0368
Random Forest	0.0752	28.4701	0.0114	0.0366

**Table 6 sensors-24-02348-t006:** Performances test week from 17 to 23 Dember 2022 for roads included in the training.

Cluster	SensorID	MAE	SMAPE	MSE	RMSE
1	MT10a	0.0476	16.1572	0.0036	0.0108
	MT10b	0.0449	9.0797	0.0042	0.0113
	MT6a	0.0591	18.2712	0.0066	0.0134
	MT6b	0.0341	10.3941	0.0024	0.0086
	MT7a	0.0438	12.3279	0.0036	0.0121
2	MT13a	0.1084	31.3894	0.024	0.0325
	MT13b	0.0979	30.4161	0.0202	0.0305
	MT17a	0.0788	23.6740	0.0103	0.0256
3	MT14a	0.0818	34.0268	0.0105	0.0248
	MT14b	0.0424	14.8554	0.0039	0.0109
	MT18b	0.0350	18.9106	0.0028	0.0087
	MT9a	0.0676	20.1880	0.0095	0.0202
	MT9b	0.0966	40.1665	0.0193	0.0348

**Table 7 sensors-24-02348-t007:** Performances test for Christmas week for roads included in the training.

Cluster	SensorID	MAE	SMAPE	MSE	RMSE
1	MT10a	0.0647	18.9955	0.0076	0.0316
	MT10b	0.0549	12.9024	0.0077	0.0284
	MT6a	0.0691	19.9931	0.0094	0.0355
	MT6b	0.0437	14.1862	0.0044	0.0264
	MT7a	0.0685	25.1541	0.0105	0.0428
2	MT13a	0.0858	25.5183	0.0111	0.0426
	MT13b	0.0854	24.5521	0.0152	0.0448
	MT17a	0.0933	30.7933	0.0122	0.0402
3	MT14a	0.0468	29.0485	0.0068	0.0224
	MT14b	0.0774	28.9589	0.0116	0.0347
	MT18b	0.0657	29.7903	0.0122	0.0355
	MT9a	0.0752	31.8823	0.0153	0.0362
	MT9b	0.0848	48.4313	0.0166	0.0374

**Table 8 sensors-24-02348-t008:** Performance metrics on excluded time series for the week of the 17th to December 23rd.

Cluster	SensorID	MAE	SMAPE	MSE	RMSE
1	MT10a	0.0506	18.3951	0.0043	0.0116
	MT10b	0.0473	9.4515	0.0047	0.0124
	MT6a	0.0598	19.1754	0.0073	0.0145
	MT6b	0.0436	16.0013	0.0045	0.0242
	MT7a	0.0483	14.7691	0.0040	0.0115
2	MT13a	0.1184	39.1892	0.0261	0.0405
	MT13b	0.1208	35.1176	0.0315	0.0409
	MT17a	0.0883	25.5717	0.0125	0.0271
3	MT14a	0.0818	34.0268	0.0105	0.0248
	MT14b	0.0838	32.7789	0.0111	0.0233
	MT18b	0.0467	21.3571	0.0045	0.0113
	MT9a	0.0779	23.0636	0.0129	0.0273
	MT9b	0.1100	53.3995	0.0233	0.0400

**Table 9 sensors-24-02348-t009:** Performance metrics on excluded time series for the week of 24 to 31 December.

Cluster	SensorID	MAE	SMAPE	MSE	RMSE
1	MT10a	0.0631	19.8354	0.0071	0.0307
	MT10b	0.0570	13.4556	0.0077	0.0293
	MT6a	0.0676	20.1005	0.0097	0.0351
	MT6b	0.0681	23.2950	0.0102	0.0314
	MT7a	0.0741	29.0209	0.0113	0.0459
2	MT13a	0.1025	35.4457	0.0176	0.0552
	MT13b	0.0853	24.7775	0.0106	0.0434
	MT17a	0.0940	30.1814	0.0119	0.0424
3	MT14a	0.0818	34.0268	0.0105	0.0248
	MT14b	0.0883	31.5855	0.0163	0.0393
	MT18b	0.0682	40.6155	0.0138	0.0359
	MT9a	0.0716	41.6216	0.0126	0.0321
	MT9b	0.0824	58.9492	0.0159	0.0403

## Data Availability

The data and the codes presented in this study are available upon request from the corresponding author.

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
