# Peer review of "Proposal of a Machine Learning Approach for Traffic Flow Prediction"

_sensors, 2024, doi:10.3390/s24072348_

Round 1
Reviewer 1 Report
Comments and Suggestions for Authors
It is a very interesting research, based on Machine learning and applied to traffic flow modeling in Italy (Case study - Catania)... The paper represents an extremely important segment of research on the 24-hour traffic flow prediction model. The paper presents a short-term and long-term approach to forecasting. The authors approached the paper with extremely precise methodological approach. The abstract of the paper describes the problem extremely well and indicates the objectives. The authors present an excellent literature review.
Special praise to the authors for the methodology of the paper and the given case study. To make the paper even better, I give certain suggestions:
Remove from keywords " machine learning, traffic flow". These words are in the TITLE.
I suggest the authors to include the following literary source in the INTRODUCTION (or chapter 2):
Kotapati, G., Ali, M. A., & Vatambeti, R. (2023). Deep Learning-Enhanced Hybrid Fruit Fly Optimization for Intelligent Traffic Control in Smart Urban Communities. Mechatron. Intell Transp. Syst., 2(2), 89-101. https://doi.org/10.56578/mits020204
Feroz Khan, A. B. & Ivan, P. (2023). Integrating Machine Learning and Deep Learning in Smart Cities for Enhanced Traffic Congestion Management: An Empirical Review. J. Urban Dev. Manag., 2(4), 211-221. https://doi.org/10.56578/judm020404
Line 44-46 Change the sentence. It is unclear what the authors wanted to write?!
Line 51-53 “Furthermore, additional experiments were conducted to test the models during holidays such as Christmas, when there are temporal irregularities in the traffic flow” - Suggestion for changing the sentence.
Chapter 2. I suggest that the title be 2. Literary review
Line 114 “heterogeneous roads” - Do the authors mean roads with heterogeneous traffic flow?
Line 139 “Catania, situated in the eastern side of Sicily, is a city with a population of approximately 300,000 inhabitants spread across an area of approximately 183 km2” - I suggest the authors delete the sentence.
Line 512-533 and Line 555-592 - I am of the opinion that these parts are redundant in the paper. Authors decide whether to change or not to change.
Reviewer 2 Report
Comments and Suggestions for Authors
This paper underscores the vital importance of precise forecasting of traffic flow, which is recognized as a key strategy for alleviating urban traffic congestion. To tackle the challenge of traffic flow prediction, a novel two-tiered machine learning approach is proposed. The comments are as follow:
1 The motivation and innovation of the research are not clear in the Introduction, a short introduction to the proposed model can be complemented.
2 A novel two-tiered machine learning model is introduced, and the inclusion of a visual representation of the model would greatly enhance the reader's comprehension of the author's innovation.
3 The existing deep learning methods are not comprehensively discussed, and graph neural networks are omitted, Phase Space Graph Convolutional Network in ieee tii, and Phase space visibility graph in chaos solitons fractals.
4 The evaluation of the model prediction step size should be provided.
Comments on the Quality of English LanguageMinor editing of English language required
Round 2
Reviewer 2 Report
Comments and Suggestions for Authors
Most of the problems have been revisied, and some recent research can be added to the related work, https://doi.org/10.1109/TII.2024.3363089.
